# The Neuroprotective Role of Quinoa (*Chenopodium quinoa*, Wild) Supplementation in Hippocampal Morphology and Memory of Adolescent Stressed Rats

**DOI:** 10.3390/nu16030381

**Published:** 2024-01-27

**Authors:** Gonzalo Terreros, Miguel Ángel Pérez, Pablo Muñoz-LLancao, Amanda D’Espessailles, Enrique A. Martínez, Alexies Dagnino-Subiabre

**Affiliations:** 1Instituto de Ciencias de la Salud, Universidad de O’Higgins, Rancagua 8370993, Chile; gonzalo.terreros@uoh.cl (G.T.); amanda.despessailles@uoh.cl (A.D.); 2Auditory and Cognition Center (AUCO), Santiago 8320000, Chile; 3Health Sciences School, Universidad Viña del Mar, Viña del Mar 2580022, Chile; miguel.perez@uvm.cl; 4Department of Cell Biology, School of Medicine, Yale University, New Haven, CT 06510, USA; pablo.munozllancao@yale.edu; 5Foyer de Charité de Provence, 13410 Lambesc, France; emartinez@foyer-sufferchoix.fr; 6Laboratory of Stress Neurobiology, Faculty of Sciences, Institute of Physiology, Universidad de Valparaíso, Valparaíso 2360102, Chile

**Keywords:** quinoa, stress, fatty acids, hippocampus, memory

## Abstract

Brain physiology and morphology are vulnerable to chronic stress, impacting cognitive performance and behavior. However, functional compounds found in food may alleviate these alterations. White quinoa (*Chenopodium quinoa*, Wild) seeds contain a high content of n-3 fatty acids, including alpha-linolenic acid. This study aimed to evaluate the potential neuroprotective role of a quinoa-based functional food (QFF) in rats. Prepubertal male Sprague-Dawley rats were fed with rat chow or QFF (50% rat chow + 50% dehydrated quinoa seeds) and exposed or not to restraint stress protocol (2 h/day; 15 days). Four experimental groups were used: Non-stressed (rat chow), Non-stressed + QFF, Stressed (rat chow) and Stressed + QFF. Weight gain, locomotor activity (open field), anxiety (elevated plus maze, light-dark box), spatial memory (Y-maze), and dendritic length in the hippocampus were measured in all animals. QFF intake did not influence anxiety-like behaviors, while the memory of stressed rats fed with QFF improved compared to those fed with rat chow. Additionally, QFF intake mitigated the stress-induced dendritic atrophy in pyramidal neurons located in the CA3 area of the hippocampus. The results suggest that a quinoa-supplemented diet could play a protective role in the memory of chronically stressed rats.

## 1. Introduction

Stress refers to the physiological response induced in living organisms when they are exposed to adversity and try to adapt to it, a process known as allostasis [1]. Non-adaptation generates allostatic overload in the brain, increasing vulnerability to mental disorders [2]. In animal models, allostatic overload triggers morphological changes in neurons, such as dendritic atrophy and neurogenesis reduction in the hippocampus, dendritic arborization in the amygdala, and dendritic remodeling in the medial prefrontal cortex [3,4]. These morphological changes impact synaptic function, leading to cognitive and behavioral impairments [5,6]. The hippocampus, a brain area associated with episodic memory, is highly vulnerable to allostatic overload. Chronic stress induces dendritic atrophy in the hippocampal CA3 pyramidal neurons [5,7] and dendritic hypertrophy in the basolateral amygdala, both leading to spatial memory impairments [4,8] and enhancing anxiety [9], respectively.

The adolescent brain is highly responsive and susceptible to environmental demands, such as stressors [10]. For instance, adolescent rats exposed to restraint stress exhibited higher plasma corticosterone levels than adult rats [11]. The serum corticosterone levels of acutely stressed adolescent rats remained elevated for twice as long as those of adult rats subjected to the same conditions [12]. In humans, both male and female adolescents also demonstrate an increased cortisol response to stress exposure compared to male and female children and adults [13].

External factors, such as nutrition and exercise, can modulate vulnerability to stress [14]. In line with this, polyunsaturated fatty acids (PUFAs) play an essential role in brain development [15] and promote anti-stress effects in rats [14]. *n*-3 PUFA supplementation improves memory and GABAergic activity in the hippocampus, modulated by the cannabinoid receptor type 1 [14]. An optimal *n*-6:*n*-3 ratio (1:5) of PUFA promotes anti-inflammatory and antioxidant activities, as has been shown in animal models [16] and in humans [17]. Moreover, a high *n*-6:*n*-3 ratio is associated with depression symptoms, a stress-related disorder [18]. As *n*-3 PUFA consumption has decreased in recent years, triggering an imbalance in the *n*-6:*n*-3 ratio, it is crucial to promote and study foods high in these fatty acids [19].

Quinoa (*Chenopodium quinoa*, Wild) is an ancient Andean grain with exceptional nutritional value. Quinoa has a high content of unsaturated fatty acids, constituting 88% of the total fatty acids, primarily oleic (OA, 18:1 *n*-9, 19.7–29.5%), linoleic (LA: 18:2 *n*-6, 49.0–56.4%) and alpha-linolenic (ALA: 18:3 *n*-3, 8.7–11.7%) acids [20]. ALA and LA are essential fatty acids, and dietary intake serves as a source of PUFAs [21]. Quinoa consumption has been associated with improved cardiovascular health and reduction of risk factors [22]. Recently, a red-quinoa seed extract prevented memory deficit induced by scopolamine in mice, suggesting that quinoa may have a neuroprotective effect [23]. Considering that hippocampal neurogenesis and maturation in the adolescent brain are particularly susceptible to stress, we aimed to evaluate the effects of a quinoa-based functional food intake in a repeated restraint stress protocol in adolescent rats. Anxiety, spatial memory, and dendritic length in long-shaft pyramidal neurons of the CA3 hippocampal region were measured in rats supplemented (or not) with quinoa-based functional food and exposed (or not) to a restraint stress.

## 2. Materials and Methods

### 2.1. Animals and Restraint Protocols

Prepubertal male Sprague-Dawley rats, 21 postnatal days (PND) old, were housed in groups of three on a 12 h light/dark cycle (light: 350 lux at 8:00 a.m.). All rats had ad libitum access to food and water at 21 ± 1 °C and 55% humidity.

Table 1 shows the experimental groups and the number of rats used in each group. Non-stressed animals, littermates of the Stressed group animals, were housed separately and not subjected to any experimental stress. Stressed groups were restrained during the light phase of the cycle at 10 a.m. Rats were placed in a custom-made plastic restrainer (12 cm long and 6 cm diameter ×20 cm, with length adjusted for growth) in their home cages for 2 h daily, from 36 to 51 PND (15 days). This protocol adapts the repeated restraint paradigm used by McLaughlin et. al. [24]. Rats were fed with commercial rat chow (Champion^®^, Rancagua, Chile) or quinoa functional food (QFF) for 31 or 32 days starting on the weaning day (day 21). The stress protocol was initiated on PND 36 (15 days after weaning) (Figure 1). To monitor the general effects of the stress protocol, we measured the weight gain in all of the animals throughout the experiments.

### 2.2. Quinoa Functional Food: Raw Material and Formulation

The quinoa-supplemented rat food for QFF groups was prepared by mixing commercial rat chow (Champion S.A.^®^, Rancagua, Chile) and quinoa in a 1:1 ratio, following previously described methods [25]. Quinoa seeds were obtained from a crop grown in coastal central Chile at Cahuil, Region of O’Higgins. Quinoa seeds were washed under stirring in water for 1 h to extract saponins. Subsequently, 125 g of quinoa was blanched in 250 mL of water at 80 °C for 15 min. Finally, quinoa was combined with an equal amount of rat chow (125 g each) and 300 mL of water. The mixture was homogenized using a Meat Mincer (TJ12 model, Australia) and molded into a pellet, similar to commercial rat chow. The pellets were dried for 36 h at 50 °C and 60% humidity in an industrial furnace.

### 2.3. Behavioral Procedures

Tests were performed between 8:00 am and 12:00 pm on naive rats recorded by cameras connected to a computer in an adjacent room. Videos were acquired by NUUO SCB-IP+ 16 software (Version 3.3.0) (NUUO, Miaoli City, Miaoli Country, Taiwan, China) and analyzed offline using EthoVision^®^ XT 18 version (Noldus, Wageningen, The Netherlands). Mazes were cleaned with 5% ethanol solution between trials. Animals from all experimental groups were evaluated simultaneously in a sound-proof and temperature-controlled (21 ± 1 °C) room. The background noise level in the room was 40 dB (Precision sound level meter, Quest Technologies, Medley, FL, USA).

(a)Open Field (OF)

Locomotor activity was analyzed 24 h after the last stress session. The open field arena (70 × 70 × 40 cm) was digitally divided into 16 grids using the EthoVision^®^ XT 18 version (Noldus, Wageningen, The Netherlands), and it was illuminated to 300 lux. The open field test was performed by placing the animal in the center of the cage for 5 min. Total distance traveled and time spent in the center were measured as locomotor and anxiety-like behaviors, respectively. Immediately afterward, anxiety levels were measured using elevated plus maze and light-dark box tests.

(b)Elevated plus maze (EPM)

The elevated plus maze consisted of two open (60 × 15 cm) and two closed (60 × 15 × 20 cm) arms arranged opposite each other extending from a central platform (15 × 15 cm). It was elevated 100 cm and illuminated by a ceiling bulb giving 300 ± 10 lux in open arms and 210 ± 10 lux in closed arms. Each rat was placed at the center of the maze, always facing the same open arm, and the frequency of entries into the open and closed arms during a 5 min test period was recorded. Entries into an arm were defined as occurring when the animal placed 70% of its body onto the arm. Open-arm entries were used as a measure of anxiety level.

(c)Light Dark Box (LDB) paradigm

Since the LDB paradigm has a different sensibility than EPM in evaluating anxiety levels [26], a new group of rats was used to perform this test. The LDB consisted of a two-compartment Plexiglass box, each measuring 50 cm × 50 cm × 40 cm each. The lighted chamber was illuminated from above by a white light bulb (500 lux at floor level), while the dark chamber was made with black Plexiglas with a light intensity of 5 lux on the floor. Both chambers were separated by a black partition with a small opening (8 cm × 8 cm) at the bottom. Each rat was placed in the center of the lighted box, facing away from the door, and released during the 5 min test. The frequency of full-body entries into the light side was measured by using EthoVision^®^ XT 18 version (Noldus, Wageningen, The Netherlands).

(d)Y-maze

Y-maze is a robust and well-documented test to measure spatial and working memory for chronically stressed rats [4]. A new group of rats underwent the Y-maze task 24 h after completing the evaluations for locomotor activity and anxiety. Each rat was placed in the center of a black Plexiglass cage (70 × 70 × 40 cm) for 5 min with arena illuminated at 300 ± 20 lux. One arm of the maze was designated as the “novel arm”, and the start and alternate arms were labeled as the “other arm” with assignments counterbalanced among rats. Stressed and non-stressed rats were simultaneously tested in different rooms and mazes.

The test comprised two phases. Initially, a training session was conducted where the novel arm was blocked, and the animals were placed on the start arm to explore the two free arms for 15 min. After training, rats were returned to their home cages, and the novel arm was unblocked. Four hours later, rats places back on the same start arm used during training and allowed to explore all three arms for 5 min. Entry into the arm was defined as when the animal placed all limbs or 70 percent of the body onto the arm. The number of entries in the novel and other arms was recorded.

### 2.4. Morphological Data Analysis

Hippocampal morphology was assessed in a new group of rats. Animals were euthanized under deep anesthesia 1 day after the end of the stress paradigm. The brain was extracted and processed using the FD Rapid Golgi Stain kit (FD Neuro Technologies, Inc., Columbia, MD, USA). Each rat was anaesthetized with 5% isoflurane (Novafarma Service S.A, Quilicura, Chile) in O_2_ using a gas anesthesia system (Ugo Basile, Gemonio, Italy) and then decapitated. The brain tissue was cut in the coronal plane, preserving the middle part containing the hippocampus.

Coronal sections, 150 µm thick, were obtained using a cryostat (Microm International, Walldorf, Germany) and collected onto super-frost plus slides. Subsequently, the sections were serially hydrated in a descending alcohol battery, cleared in xylene, and cover-slipped. The drawing of stained cells (long-shaft pyramidal neurons of CA3 hippocampal neurons) was generated using Camera Lucida tracking (BX31-U-DAL 10X, Olympus Co., Tokyo, Japan). Neurons were randomly selected based on the following criteria: (1) presence of untruncated dendrites, (2) consistent and dark impregnation along the entire dendritic field, and (3) relative isolation from neighboring impregnated neurons. The drawn cells were scanned (eight-bit grayscale TIFF images with 1200 d.p.i. resolution; EPSON ES-1000C, Torrance, CA, USA) along with a calibrated scale for subsequent computerized image analysis. A researcher conducted data analysis (dendritic arbor length per neuron) in a double-blind fashion. ImageJ 1.47 software (Wayne Rasband, National Institute of Health, Bethesda, MA, USA) was used for the morphometric analysis of digitized images.

### 2.5. Statistical Analysis

First, all variables were analyzed for normal distribution using the Shapiro–Wilk test, whereas the homoscedasticity was assessed using the Levene test. Weight gain differences were analyzed by two-way ANOVA [groups (stress, diet) × days (1, 5, 10, 15)] followed by Bonferroni post-test. Results from the OF (total distance traveled, time spent in the center), EPM (percentage of open-arm entries), LDB (light side entries), and Y-maze tests (total entries into the novel arm) and dendritic morphology were analyzed by two-way ANOVA for stress (non-stressed or stressed) and diet (rat chow or QFF), followed by a Bonferroni post-test. Statistical analyses were performed using Prism 7 software (GraphPad Software Inc., La Jolla, CA, USA). Results are presented as mean ± SEM, and a probability level of 0.05 or less was accepted as significant.

## 3. Results

### 3.1. QFF Intake Did Not Affect Body Weight Gain in Rats Subjected to a Restraint Stress

To assess the effect of the stress protocol and QFF intake, we measured the body weight gain during the stress protocol. Figure 2A displays the body weight gain over the 15 days of the stress protocol. Rats under the stress protocol exhibited significantly lower weight gain compared to the non-stressed rats (*p* < 0.0001). Conversely, the QFF diet did not affect body weight gain compared to chow diet (*p* > 0.05). Figure 2B displays the reduction in total weight gain in the stressed group at the end of the 15-day protocol (*F*_(1.31)_ = 42.94, *p* < 0.0001), with no significant differences in body weight gain observed for diet. This suggests that QFF intake did not alter body weight gain and did not prevent the stress-induced effects on this parameter (Figure 2B).

### 3.2. QFF Consumption Did Not Affect Locomotor Activity and Anxiety-like Behavior

We measured the locomotor activity in the OF test and anxiety-like behavior in the OF, EPM, and LDB tests 24 h after the last stress session. Figure 3A shows that the locomotor activity was not affected by stress protocol or diet (*p* > 0.05). Similarly, the analysis of the anxiety-like behavior indicator in the OF test, measured as the time spent in the center of the field (Figure 3B), was not affected by stress protocol or diet (F_(1.32)_ = 0.1028, *p* = 0.7506). To complement the OF test results, we analyzed the anxiety levels in the EPM and LDB tests at PND 52. Figure 4A shows that the restraint stress reduced the number of open-arm entries in the EPM, confirming that our chronic stress protocol induced an anxiety-like behavior (Figure 4A). There was no main effect of diet (F_(1.32)_ = 1.2, *p* = 0.2784) or diet–treatment interaction (F_(1.32)_ = 0.065, *p* = 0.8008) on anxiety levels measured in the EPM (Figure 4A). We next assessed anxiety in the LDB (Figure 4B), where rats subjected to the stress protocol showed a significant decrease in the number of entries to the light side compared to the non-stressed groups (F_(1.32)_= 14.80, *p* < 0.001). There were no main effects of diet or interaction (diet–stress) on the numbers of entries to the light side (Figure 4B). These results showed that quinoa did not affect either locomotor activities or anxiety behavior in our model.

### 3.3. QFF Consumption Prevented Spatial Memory Impairment Induced by Restraint Stress

To determine whether QFF consumption could modulate cognitive parameters in our stress protocol, we evaluated spatial memory impairments in the animals using the Y-maze task (Figure 5) 48 h after the last stress session. Restraint stress significantly decreased the number of entries made into the novel arms of the Y-maze (*p* < 0.05). There was not a main effect of diet on the number of entries made into the novel arms of the Y-maze. However, there was a main interaction effect (diet × tress) on the number of entries made into the novel arms of the Y-maze (F_(1.49)_ = 15, *p* = 0.0004). Further analysis showed that QFF consumption prevented the memory deficit induced by stress (*p* < 0.05; Stressed (rat chow) vs. Stressed + QFF entries to the novel arm), allowing us to conclude that quinoa consumption was effective in modulating spatial memory in our model of stress.

### 3.4. QFF Consumption Reduced the Stress-Induced Dendritic Atrophy in the Hippocampus

To determine whether the effect of QFF consumption on spatial memory was associated with morphological changes in the hippocampus, we compared the total dendritic length of CA3 pyramidal neurons of rats from non-stressed and stressed groups. First, the Bonferroni–Dunn post-test analysis revealed that the stress protocol decreased the total dendritic length compared to that of non-stressed animals (*p* < 0.05) (Figure 6). Second, QFF consumption effectively increased the total dendritic length to the values observed in non-stressed animals (stressed 1544 ± 486.3 µm, stressed + QFF 2338 ± 1107 µm) (Figure 6). Moreover, QFF supplementation in the non-stressed group did not have a significant effect (*p* > 0.05) on the dendritic length of the CA3 pyramidal neurons. These results suggest that quinoa supplementation prevented the dendritic atrophy in hippocampal neurons in our stress model.

## 4. Discussion

In this study, we demonstrate that the consumption of quinoa-based food during adolescence enhances the memory of stressed rats and mitigates stress-induced dendritic atrophy in the hippocampus. Initially, we examined whether our stress protocol effectively elicited stress responses. Stressed rats from all experimental groups (rat chow and QFF) exhibited less body weight gain than non-stressed rats. This indicates the effectiveness of the stress protocol, and the intake of QFF did not prevent the reduction in weight gain induced by restraint stress. However, other studies have reported that the dietary supplementation of hydrolyzed quinoa (2000 mg/kg, 30 days) in sedentary Wistar rats resulted in decreased body weight gain, food intake, fat deposition, and triglycerides [27]. Additionally, supplementation with fermented or sprouted quinoa (47 days) reduced food intake, blood glucose, and lipid levels in rats fed with a high-carbohydrate diet [28]. Both studies reported a decrease in weight gain after quinoa supplementation, which may be associated with reduced food intake [28] linked to lower levels of ghrelin, leptin, and cholecystokinin [29].

On the other hand, we did not observe changes in the body weight gain in the non-stressed group fed with QFF compared to rat chow, which could be explained by differences in food intake, forms of administration, and nutritional properties of the food. We suggest that the administration of anxiogenic protocols (oral administration or orogastric gavage) and reduction in food intake result in less body weight gain. In support of this idea, previous studies have shown that oral administration of *n*-3 PUFAs is considered a stressor for rats because it increases corticosterone levels in non-stressed rats [30], thereby reducing body weight gain. Moreover, feeding the animals with a solution of hydrolyzed quinoa through orogastric gavage reduces food intake and decreases body weight gain [27].

Food intake is strongly influenced by stress and the composition of the diet. Rats subjected to stress consumed less commercial chow than the non-stressed rats [31]. Interestingly, rats given the option to choose between two diets prefer high-caloric food [31]. In our study, we found no difference in food intake. This may suggest that our method of administration of food is less stressful or that, due to nutritional characteristics, quinoa is more palatable for rats.

Chronic stress-induced anxiety arises from alterations in neuronal morphology in the basolateral amygdala [32]. In the brain, the medial prefrontal cortex and the amygdala are extensively interconnected, working together to regulate the expression of emotions such as fear and anxiety [33]. It is possible that the chronic stress protocol used in our study induced hyperactivation of the basolateral amygdala and/or the bed nucleus of the stria terminalis [33,34], leading to increased anxiety. On the contrary, it has been demonstrated that supplementation with *n*-3 PUFAs can reduce chronic stress-induced anxiety [30,35]. The anxiolytic effect of *n*-3 PUFAs was notably different from controls only in subgroups with a higher dosage (at least 2000 mg/d) and not in subgroups with a lower dosage (<2000 mg/d) [36]. This suggests that the anxiolytic effect depends on the concentration of PUFA, specifically docosahexaenoic acid (DHA: 22:6 *n*-3) that is synthesized from ALA. In our experiments, QFF consumption did not affect anxiety in the EPM and LDB tests. Quinoa seeds are rich in ALA [37], but while humans can convert ALA into eicosapentaenoic acid (EPA: 20:5 *n*-3), which is the main substrate for synthesizing DHA, the efficiency of this conversion is limited, and thus EPA and DHA must also be obtained from the diet to maintain adequate levels [38]. Therefore, we believe that due to the low concentration of DHA achieved from the conversion of ALA, it is not possible to obtain an anxiolytic effect.

The chronic restraint protocol induces atrophy in dendritic arborization of pyramidal neurons from the CA3 region of the hippocampus, characterized by a reduction in dendritic length and number of branch points [39]. Moreover, dendrite atrophy has been associated with deficits in hippocampal-dependent spatial memory [8,39]. This evidence aligns with our findings, where the restraint protocol led to a decrease in dendritic length and spatial memory deficits assessed in the Y-maze. However, QFF intake in our study mitigated these effects, alleviating the stress-induced damage in both dendritic length and spatial memory. Notably, prior studies have demonstrated that an extract from red quinoa seeds prevented memory deficits induced by scopolamine treatment in mice [23]. This supports our findings and establishes the protective role of a quinoa-enriched diet in cognitive performance and/or behavioral alterations in rodents.

Diet significantly influences anxiety behavior [40], learning, and memory in rats [41]. When combined with stress, a diet rich in saturated fatty acids induces the retraction of dendrites in neurons in the CA3 region of the hippocampus, leading to arterial hypertension and an increase in corticosterone levels [14,42]. In contrast, *n*-6 and *n*-3 PUFA consumption improves the cognitive and emotional state in both rodents [43] and humans [44]. Quinoa seeds have a high content of ALA and LA, two essential fatty acids, which represent 55–63% of its lipid fraction [45]. ALA is associated with health, reproduction, and mammalian development [46]. Importantly, ALA can be converted to long-chain PUFAs (LCPUFAs) in the liver, enabling the synthesis of EPA and DHA. DHA, the predominant fatty acid in the brain, constituting about 15% of the total fatty acids in that tissue [47], plays a crucial role. Inadequate DHA intake has been linked to poor performance in attention and learning tests [48], increased aggression, and elevated anxiety levels [49] in rodents. Additionally, rats deficient in *n*-3 LCPUFAs exhibit excessive stress, anxiety, and fear responses, which are reversed after DHA supplementation [50]. Moreover, EPA and DHA supplementation improved memory and prevented hippocampal dendritic atrophy in chronically stressed rats [51]. DHA may prevent stress-induced behavior through modulating GABA receptor activity [50] and restoring GABA release probability in the CA1 region of the hippocampus [51]. Conversely, the consumption of a fish oil-enriched diet (high in EPA and DHA) alters the phospholipid composition of the rat brain, increasing levels of phosphatidylserine and influencing cortical and striatal dopaminergic function. This dietary change enhances cognitive function, spatial memory, and locomotor activity in aged rats [52]. Furthermore, *n*-3 PUFA induces the expression of genes involved in learning, memory, and neuronal metabolism [53]. Interestingly, high ALA supplementation in rats did not alter anxiety levels [52,54], aligning with our findings that QFF did not affect anxiety in the EPM and LDB tests.

In summary, the collective evidence suggests that QFF’s effects in preventing spatial memory decline and dendritic impairment induced by the restraint protocol may be attributed to the PUFA content in quinoa. However, the observed differences between anxiety and memory effects could be attributed to varying requirements in the concentration of PUFAs, emphasizing the need for further studies.

In addition to the lipid profile of quinoa, it has been suggested that its high antioxidant content may play a role in the neuroprotective effects of these seeds. Indeed, dietary supplementation with red quinoa extract, rich in phenolic and flavonoid compounds, demonstrated a neuroprotective effect in mice treated with scopolamine [23].

## 5. Conclusions

We presented evidence supporting the neuroprotective effects of a quinoa-supplemented diet on neuronal morphology and memory. Additionally, we found that quinoa did not alleviate anxiogenic behavior or changes in weight gain induced by stress. This study enhances our understanding of the positive impact of quinoa on brain physiology. Future research should investigate whether quinoa’s effects on behavior are primarily due to the ALA content or other nutrients found in various quinoa seeds. Ultimately, considering its potential, this amaranthaceous plant could be considered a healthy dietary option to mitigate the impact of toxic stress (allostatic overload) in humans. For example, in stress-related disorders where memory deteriorates, such as in Alzheimer’s disease, dietary interventions using quinoa could be an alternative to enhance the therapeutic effect of drugs currently available for these neurodegenerative diseases.

## Figures and Tables

**Figure 1 nutrients-16-00381-f001:**
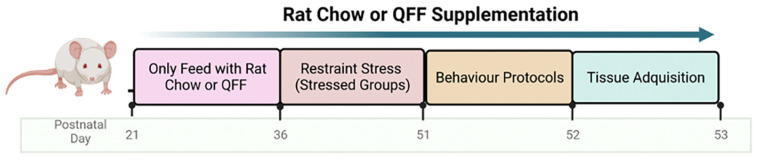
Schematic drawing of the experimental design. Male Sprague-Dawley rats of 21 postnatal days (PND) were fed either with rat chow or quinoa functional food (QFF, (50% rat chow + 50% dehydrated quinoa seeds)). At PND 36, animals were submitted to a restraint protocol for 2 h per day for 15 days, establishing four groups (*n* = 27 per experimental group): non-stressed (rat chow), non-stressed + QFF, stressed, and stressed + QFF. At PND 52, motor and anxiety behavior were evaluated on the open field (OF), elevated plus maze (EPM), and light-dark box (LDB). Finally, at PND 53, the memory performance was evaluated on the Y-maze test, and brain tissue was used for Golgi staining. Created with BioRender.com.

**Figure 2 nutrients-16-00381-f002:**
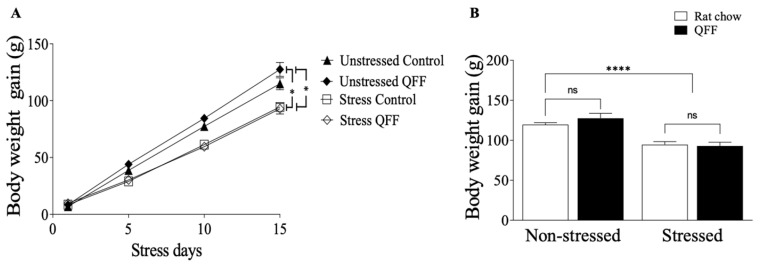
Effect of diet type on body weight gain during the stress protocol: (**A**) Daily measure of body weight during the 15 days of stress protocol, and (**B**) total body weight gain at the end of day 15 of the stress protocol. There were no significant differences in body weight gain induced by diet intake. However, exposure to the restraint stress protocol was associated with a significant reduction in body weight in both rat chow and QFF groups (*n* = 9, in each). Values are expressed as mean ± SEM (*n* = 9). * *p* < 0.05, **** *p* < 0.0001. Non-significant, n.s; quinoa functional food, QFF.

**Figure 3 nutrients-16-00381-f003:**
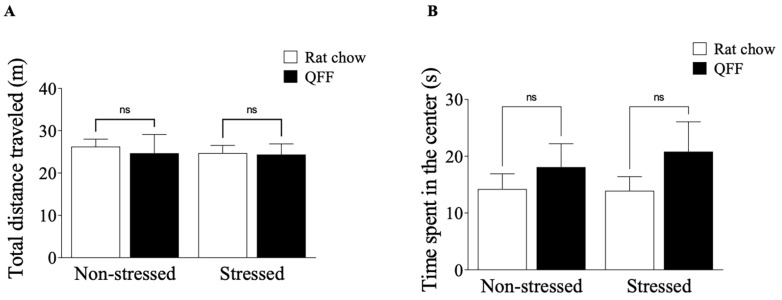
Effect of diet type on locomotor activity measured in the open field test. (**A**) Total distance traveled (m) and (**B**) time spent in the center in an open-field test (s). Locomotor activity at PND 52 was not affected by the restraint stress protocol in any experimental group. The anxiety-like behavior measured by the time spent in the center of the field did not show a significant difference in any of the groups (*n* = 9, in each). Values are expressed as mean ± SEM (*n* = 9). Non-significant, n.s; quinoa functional food, QFF.

**Figure 4 nutrients-16-00381-f004:**
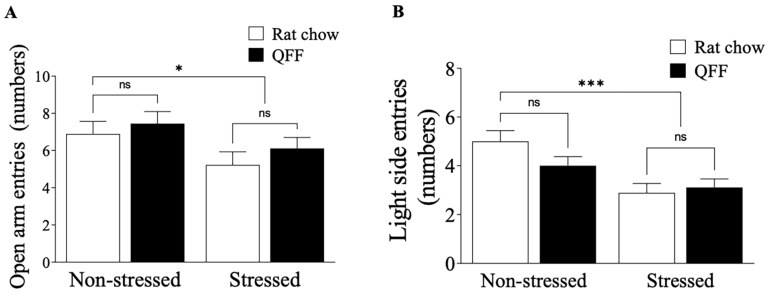
Effect of diet type on anxiety-like behavior measured in the elevated plus-maze and light-dark box test. (**A**) Total number of open arm entries made in the elevated plus-maze. Stress protocol significantly decreased the number of open-arm entries in both the rat chow and QFF groups. Anxiety levels measured in the EPM were not affected by diet. (**B**) Total number of light side entries made in the light-dark box test. Stressed animals had a smaller number of entries to the light side compared to the non-stressed animals in both the rat chow and QFF groups. Anxiety levels measured in LDB were not affected by diet between groups (*n* = 9, in each). Values are expressed as mean ± SEM (*n* = 9) * *p* < 0.05, *** *p* < 0.001. Non-significant, n.s; quinoa functional food, QFF.

**Figure 5 nutrients-16-00381-f005:**
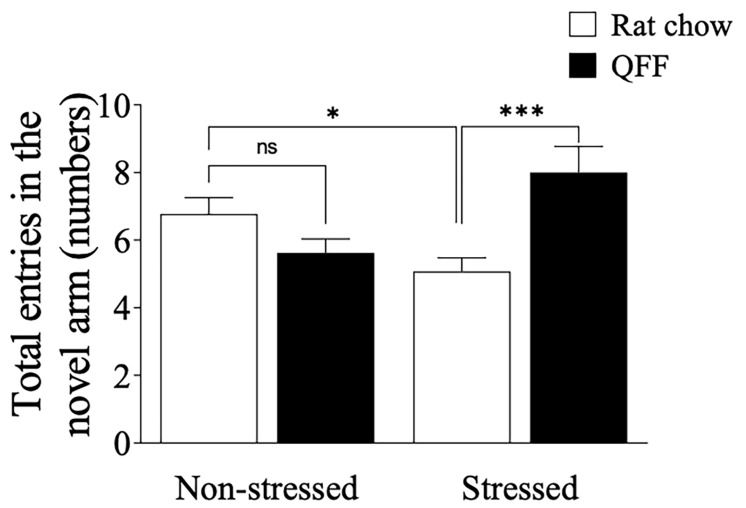
Influence of quinoa functional food diet on Y-maze performance. Total number of entries made in the novel arm in the Y-maze. Chronic stress protocol induced spatial memory impairment in the rat chow but not in the QFF group (*n* = 9, in each). Values are expressed as mean ± SEM (*n* = 9), * *p* < 0.05, *** *p* < 0.001. Non-significant, n.s; quinoa functional food, QFF.

**Figure 6 nutrients-16-00381-f006:**
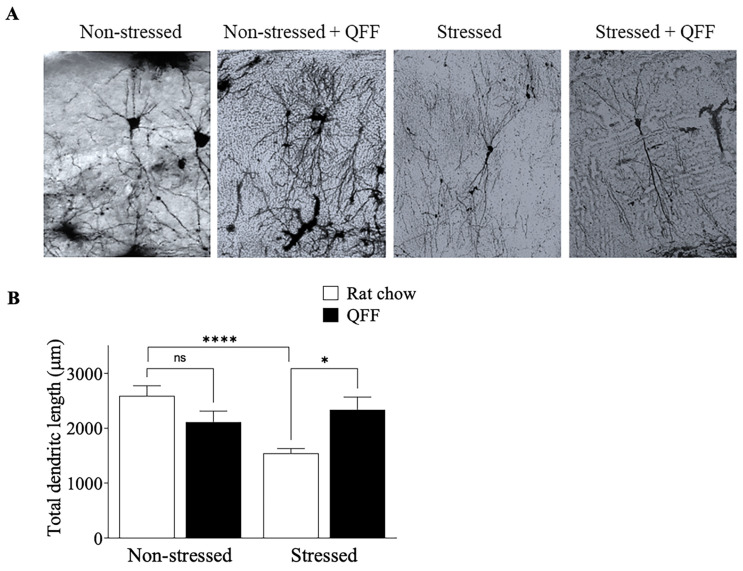
Effect of quinoa functional food on CA3 dendritic morphology. (**A**) Representative photographs and (**B**) quantification of the total dendritic length of CA3 pyramidal neurons in the hippocampus. QFF diet reduced the dendritic impairment induced by stress compared to the rat chow group (*n* = 9, in each). Values are expressed as mean ± SEM (*n* = 9), * *p* < 0.05, **** *p* < 0.0001. Non-significant, n.s; quinoa functional food, QFF.

**Table 1 nutrients-16-00381-t001:** Number of rats used in each experiment.

Description	Experimental Groups				Total
	Non-Stressed	Non-Stressed + QFF	Stressed	Stressed + QFF	
Locomotor activity and anxiety	*n* = 9	*n* = 9	*n* = 9	*n* = 9	
Golgi staining	*n* = 9	*n* = 9	*n* = 9	*n* = 9	
Behavioral tasks	*n* = 9	*n* = 9	*n* = 9	*n* = 9	
**Total**	*n* = 27	*n* = 27	*n* = 27	*n* = 27	*n* = 108

## Data Availability

The data that support the findings of this study are available from the corresponding author, A.D.-S., upon reasonable request.

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
