# Peer review of "The Neuroprotective Role of Quinoa (Chenopodium quinoa, Wild) Supplementation in Hippocampal Morphology and Memory of Adolescent Stressed Rats"

_nutrients, 2024, doi:10.3390/nu16030381_

Round 1

Reviewer 1 Report

Comments and Suggestions for Authors

Authors of manuscript show in their work that Quinoa supplementation of rat food may prevent cognitive problems associated with stress. And also confirmed by histology of hippocampus dendrites. 

Before going to publication phase, I want address few questions

1) Please describe euthanasia procedure before histology in details. What method was used, in anaesthetics were used, indicate name and doses.

2) For elevated plus maze number of transitions between zones not always best parameter to check. May you indicate time spent in open and closed arms of EPM. With Noldus Ethovision its usually easy to extract this data from recordings 

3) Same for Light Dark box, what was actual time spent in light zone by mice?

4) If it will be some difference in time spent in different zones, it can be informative to illustrate figures by cumulative plots of mice position in open field and elevated plus maze 

Comments on the Quality of English Language

Please check for some minor typos. For example in Figure one, "Tissue Acquisition" instead "adquisition"

Author Response

1) Please describe euthanasia procedure before histology in details. What method was used, in anaesthetics were used, indicate name and doses.

Answer:

            The information was added to the manuscript.

1) Please describe euthanasia procedure before histology in details. What method was used, in anaesthetics were used, indicate name and doses.

Answer:

            The information was added to the manuscript.

2) For elevated plus maze number of transitions between zones not always best parameter to check. May you indicate time spent in open and closed arms of EPM. With Noldus Ethovision its usually easy to extract this data from recordings

Answer:

As requested, we analyzed time spent in open and closed arms, but we did not find significantly differences between the groups. However, a tendency was found (open arms: non-stressed rat chow= 58.93 ± 25.79 s, non-stressed QFF=49.18 ± 27.41 s; stressed rat chow= 48.01 ± 17.22 s, stressed QFF= 42.22±20.51 s; F (1, 32) = 1.3, P=0.25; Closed arms: non-stressed rat chow= 172.2 ± 29.79 s, non-stressed QFF= 196.8 ± 71.36 s; stressed rat chow= 176.5 ± 22.68 s, stressed QFF= 184.3 ± 25.60 s; F (1, 32) = 0.085, P=0.7727). In this sense, it has been described that both percentage of entries and time spent in the open arms are indicators of anxiety (Walf and Frye, 2007).

Walf, A., Frye, C. The use of the elevated plus maze as an assay of anxiety-related behavior in rodents. Nat Protoc 2, 322–328 (2007). https://doi.org/10.1038/nprot.2007.44

3) Same for Light Dark box, what was actual time spent in light zone by mice?

Answer:

Regretfully, we did not save the data about the time spent in the light zone in the Light Dark box, so we don’t have access to it.

4) If it will be some difference in time spent in different zones, it can be informative to illustrate figures by cumulative plots of mice position in open field and elevated plus maze.

Answer:

As we did not find any effect of QFF in the time spent in both open and closed arms of the EPM, we think that a cumulative plot of the time spent in the different zones in the EPM would not add information about the experiments.

Reviewer 2 Report

Comments and Suggestions for Authors

nutrients-2740440: “The neuroprotective role of Quinoa (Chenopodium quinoa, Wild) supplementation on hippocampal morphology and memory of adolescent stressed rats”.

In this study, the authors demonstrate a protective potential of a remedy prepared from quinoa seeds on adolescent stressed rats. The material is described successively and conclusions are supported by obtained data.

A few technical and grammatical remarks:

1)     in lines 58-59, the sentence should be rewritten, e.g., “An optimal n-6:n-3 ratio (1:5) of PUFA promotes anti-inflammatory and antioxidant activities, that has been shown in animal models [16] and in humans [17].”;

2)     in lines 81 and 86, “a.m.” and “a.m..” would be correct;

3)     in line 102, “QFF, Quinoa functional food.” should be removed as this is duplicate of that denoted in line 96;

4)     in line 115, “a.m..” would be correct;

5)     in line 125, “   twenty four hours   “;

6)     in lines 170,171, the sentence should be rewritten, e.g., “Animals were killed under deep anesthesia one day after the end of the stress paradigm.”

7)     in line 176, “   serially, hydrated   “;

8)     in lines 184, 185, the sentence should be rewritten;

9)     in lines 214,242,251,268,274, “   (n=9, in each)   ” should be placed immediately after “groups” or “group”;

10)  in legends to Figures 2,4,5,6, “*** p<0.001” and “**** p <0.0001” should be added, when appropriate;

11)  in lines from 219 to 228, all numerical data should be removed as they duplicate those on Figures 3 and 4;

12)  the same remark is for the next Figures below;

13)  in lines 307-310, the sentence should be rewritten;

14)  in lines 329 and 331, “(DHA: 22:6 n-3)” and “(EPA: 20:5n-3)” need more details;

15)  in line 347, “A diet   “;

16)  The Reference list needs corrections: a) a year in bold fonts, b) volumes and pages should be added, when appropriate;

17)  English should be double checked.

Comments on the Quality of English Language

Minor editing of English language required

Author Response

1) In lines 58-59, the sentence should be rewritten, e.g., “An optimal n-6:n-3 ratio (1:5) of PUFA promotes anti-inflammatory and antioxidant activities, that has been shown in animal models [16] and in humans [17].”;

Answer:

The sentence was corrected

2) In lines 81 and 86, “a.m.” and “a.m..” would be correct;

Answer:

The lines were corrected.

3) In line 102, “QFF, Quinoa functional food.” should be removed as this is duplicate of that denoted in line 96;

Answer:

The line was corrected.

4) In line 115, “a.m..” would be correct;

Answer:

The line was corrected.

5) In line 125, “twenty four hours“;

Answer:

The line was corrected.

6) In lines 170,171, the sentence should be rewritten, e.g.,“Animals were killed under deep anesthesia one day after theend of the stress paradigm.”

Answer:

The lines were corrected.

7) In line 176, “serially, hydrated“;

Answer:

The line was corrected.

8) In lines 184, 185, the sentence should be rewritten;

Answer:

The lines were corrected.

9) In lines 214,242,251,268,274, “(n=9, in each)” should be placed immediately after “groups” or “group”;

Answer:

The lines were corrected.

10) In legends to Figures 2,4,5,6, “*** p<0.001” and “**** p<0.0001” should be added, when appropriate;

Answer:

The lines were corrected.

11) In lines from 219 to 228, all numerical data should be removed as they duplicate those on Figures 3 and 4;

Answer:

The lines were corrected.

12) The same remark is for the next Figures below;

Answer:

The lines were corrected.

13) In lines 307-310, the sentence should be rewritten;

Answer:

The lines were corrected.

14) in lines 329 and 331, “(DHA: 22:6 n-3)” and “(EPA: 20:5n-3)” need more details;

Answer:

The lines were corrected.

15) In line 347, “A diet“;

Answer:

The line was corrected.

16) The Reference list needs corrections: a) a year in bold fonts, b)volumes and pages should be added, when appropriate;

Answer:

The reference list was corrected.

17) English should be double checked.

Answer:

English was corrected.

Reviewer 3 Report

Comments and Suggestions for Authors

The study offers valuable insights into the potential neuroprotective effects of quinoa, which is relevant to the journal's scope. The study explores the effects of a quinoa-based diet on rats exposed to chronic stress. The study investigates various parameters including weight gain, locomotor activity, anxiety, spatial memory, and hippocampal dendritic morphology. Key findings suggest that while quinoa supplementation doesn't significantly alter anxiety-like behaviors or weight gain, it does improve the memory of stressed rats and counteracts stress-induced dendritic atrophy in hippocampal neurons.

Expanding the discussion to place these findings within the broader context of existing research on dietary interventions for stress-related neural changes would be beneficial. It's important to discuss the potential implications for human dietary practices and public health, while acknowledging the limitations of animal models.

It is well-written. 

minor comment:

Page 13, Line 469: "seriallyydrated" to "serially hydrated".

Author Response

1) Expanding the discussion to place these findings within the broad ercontext of existing research on dietary interventions for stress-related neural changes would be beneficial. It's important to discuss the potential implications for human dietary practices and public health, while acknowledging the limitations of animal models.

Answer:

A paragraph was included at the end of the discussion section related to the potential that this preclinical research could have to complement current treatment of stress-related diseases that affect memory.

2) Minor comment: Page 13, Line 469: "seriallyydrated" to "serially hydrated"..

Answer:

Page 13, Line 469 was corrected.